# Mode of Action of Heat Shock Protein (HSP) Inhibitors against Viruses through Host HSP and Virus Interactions

**DOI:** 10.3390/genes14040792

**Published:** 2023-03-25

**Authors:** Shuang Wu, Yongtian Zhao, Delu Wang, Zhuo Chen

**Affiliations:** 1Key Laboratory of Green Pesticide and Agricultural Bioengineering, Ministry of Education, Guizhou University, Guiyang 550025, China; 2College of Life Science and Agriculture, Qiannan Normal University for Nationalities, Duyun 558000, China; 3College of Forestry, Guizhou University, Guiyang 550025, China

**Keywords:** heat shock proteins, virus, interaction, molecular mechanism, HSP inhibitors, antiviral activity

## Abstract

Misfolded proteins after stress-induced denaturation can regain their functions through correct re-folding with the aid of molecular chaperones. As a molecular chaperone, heat shock proteins (HSPs) can help client proteins fold correctly. During viral infection, HSPs are involved with replication, movement, assembly, disassembly, subcellular localization, and transport of the virus via the formation of macromolecular protein complexes, such as the viral replicase complex. Recent studies have indicated that HSP inhibitors can inhibit viral replication by interfering with the interaction of the virus with the HSP. In this review, we describe the function and classification of HSPs, the transcriptional mechanism of HSPs promoted by heat shock factors (HSFs), discuss the interaction between HSPs and viruses, and the mode of action of HSP inhibitors at two aspects of inhibiting the expression of HSPs and targeting the HSPs, and elaborate their potential use as antiviral agents.

## 1. Introduction

Because of the constraints imposed by a small genome and its limited coding capacity, a virus depends on host factors to multiply inside host cells. As a consequence, viruses causing diseases in humans, animals, and plants are difficult to curate and control [1]. Heat shock proteins (HSPs) are specific types of proteins, which contribute to many biological functions. HSPs are induced or changed in response to virus infection, and are involved with the viral replicase complex (VRC) or other viral macromolecular protein complexes, helping the virus to complete different steps of viral multiplication, replication, translocation, assembly, disassembly, etc. Nevertheless, HSP inhibitors can regulate HSPs by various mechanisms, thus displaying different bioactivities, including anticancer and antiviral activities. In this review, we searched the literature from 1970 to 2022 by selecting “HSP”, “HSF”, “HSP inhibitor”, and “antiviral activity” as the keywords in the database of web of science and pubmed. A total of 232 papers were cited in the review. We described the functional mechanisms of interactions between HSPs and viruses. Additionally, we elaborated HSP inhibitors, which were mainly divided into two kinds: those directly acting on HSPs to affect their function, and those indirectly acting on HSFs to inhibit transcription of HSPs. Some representative HSP inhibitory compounds, such as quercetin, analogues of quercetin, flavonoid compounds, geldanamycin, etc., were used to reveal the mechanism of HSP regulation.

## 2. HSPs Are Important Molecular Chaperones

HSPs are types of proteins produced by the host organism in response to biotic and abiotic stresses which cause many proteins to aggregate and lose their function; HSPs function as molecular chaperones, disaggregating and solubilizing aggregated proteins. HSPs have been highly conserved during molecular evolution. Based on molecular weight and sequence homology, HSPs are divided into HSP100 (heat shock protein 100 kDa), HSP90, HSP70, HSP60, and small heat shock proteins (sHSPs) [2]. HSPs are located in all cellular compartments and participate in protein refolding, protein translocation, assembly of protein complexes, and the transport of proteins in an ATP-dependent manner [3,4]. In addition, HSPs, acting as intermediaries, are involved with protein degradation, disassembly of protein complexes, deactivation of damaged proteins, and regulation of many signal pathways [5]. sHSPs have the activity of heat stability through formatting polymer structure [2,6,7].

HSP70 family homologs, which include HSP70 (heat shock 70 kDa protein) and HSC70 (heat shock cognate 70 kDa protein), have a molecular weight of approximately 70 kDa. The operation of these homologs is ATP-dependent. HSP70 has a low expression level, even being undetectable, in the normal, unstressed state, but the expression is strongly induced by environmental factors, such as heat and pesticide treatments, as well as by heavy metals, oxidative stress, and viral infections [8,9,10]. HSP90s are characteristically highly abundant in cells and consist of cytoplasmic HSP90s (the stress-induced HSP90*α* and the constitutively expressed HSP90*β*), an endoplasmic reticulum (ER) resident homolog, HSP90B (GRP94), and a mitochondrial variant, TRAP1 [11,12,13]. HSP90s are homodimeric complexes with ATPase activity, and each monomer of an HSP90 possesses a common C-terminal dimerization domain, a middle domain, and a structurally unique N-terminal ATPase domain [14,15]. As an HSP90 family gene, HSP50.2 from rice can positively regulate drought stress tolerance, and possesses high potential usefulness in drought tolerance improvement of rice [16].

HSP90s from Arabidopsis coupling with the YODA (YDA) kinase pathway affect transcriptional networks, and then disturb early embryo development [17]. Members of the HSP100 subfamily form ring-like oligomers, and unfold protein aggregates to structured polypeptides, which are easily threaded through their small central channel [18]. These members are widely distributed among prokaryotes and eukaryotes, among which HSP101 plays a significant role in thermotolerance [19]. HSP101 from Arabidopsis can promote flowering under nonstress conditions [20]. sHSPs possess an antiaggregation activity towards a broad range of substrates [21].

The mechanism of recruiting and folding a client protein by HSPs always involves several steps and collaboration with other molecular chaperones. For instance, HSP70s often interact with newly synthesized proteins, whereas HSP90s often receive client proteins acted on by the HSP70 chaperone system. For instance, the steroid hormone receptors are first bound by HSP70, and then transferred by the cochaperone Hop bridges to HSP90. Specific tetratricopeptide (TPR) interaction domains, which recognize the C-terminal tails of both HSP70 and HSP90, are responsible for linking HSPs and the cochaperone Hop [22,23]. The homodimer HSP90, without a nucleotide, binds to the client protein, then HSP90 binds to ATP, triggering a conformational change of the client protein so that the client protein becomes more stable in terms of spatial conformation. Lastly, the protein complex of the client protein and HSP90 is dissociated when the nucleotide-binding HSP90 is hydrolyzed. Consequently, HSP90 reverts to the state of being nucleotide-free [24].

HSPs also use degraded proteins to participate in the maturation of recruited proteins [25]. Among them, the degradation mechanism is associated with TPR, which recognizes HSP70 and HSP90; meanwhile, the TPR can bind to E3 ubiquitin ligase, thereby initiating protein degradation of the enzyme [26].

## 3. HSFs Can Promote the Transcription of HSP Genes

Transcription of HSP genes involves heat shock factors (HSFs), which are transcriptional activators of heat shock protein genes. The regulation of HSF gene transcription has been little described, but HSF is translated and then subjected to post-translational activation by changing into a trimeric form from the monomeric form. This trimer consists of two separated subdomains with the structure of a coiled-coil [27]. HSFs bind to heat shock elements (HSEs) in the promoters of *HSP* genes to form a protein–DNA complex [28]. In many species, a characteristic sequence, “C–GAA–TTC–G”, is located within the *HSP* promoter at various distances from the site of transcription initiation [29]. Different sequences located in the *HSP* promoter confer different responses when the sequence is bound by HSF transcription factors (Figure 1) [30,31]. Lastly, HSP is phosphorylated to generate the activated form under heat shock conditions, as in the yeast *Saccharomyces cerevisiae*, or subjected to post-translational modification in Drosophila and mammalian cells, prior to binding to HSEs [32,33]. mRNAs of HSPs are ultimately transcribed from the *HSP* genes [34]. HSP expression is induced by factors such as growth or development, heat shock, or chemical inducers (metal ions, pesticides, azetidine, sodium arsenite, dinitrophenol, and sodium salicylate) [10,31,35,36]. Factors such as heat shock, heavy metal ions, and amino acid analogs interact with HSFs to regulate different processes of *HSP* gene transcription or HSP synthesis [37]. HSF3, HSF13, and HSF17 from maize can bind to several HSP20 promoters during the response to abiotic stress [38]. Transgenic HSP26.8s from *Agrostis stolonifera* L. are more susceptible to heat and salt, but less sensitive to drought stress compared to wild-type plants, and are characterized as negatively regulating plant growth containing arrested root development, slow growth rate, and twisted leaf blades [39]. Different *HSF* genes have similar structural characteristics; that is, there are DNA-binding domains and trimer-binding domains in the core structural regions (Figure 2). Based on protein sequence homology, HSFs are divided into three types, namely, A, B, and C, with each type represented by a family composed of many members. All HSFs contain a similar structure of DNA-binding motifs, an oligomerization domain, and a nuclear localization sequence. Because of differences in polypeptide sequences, the families of A and C differ with respect to activation or inhibitory activity, respectively. The HSFA family is responsible for responses to heat and other stress factors, but the HSFB and HSFC families have been little studied [40]. In tomato plants, HSFB1 and HSFA1 can regulate heat stress by forming a synergistic coactivator [41]. In cooperation with other transcription factors, HSFB1 can control and restore gene expression in response to heat shock [42].

## 4. HSPs Participate in Different Viral Functions

The viral macromolecular complexes (MCs) consist of virus-encoded proteins, host-derived proteins, the viral RNA template, nucleotides, etc. These components are assembled into, for example, the virus replication complex (VRC) on intracellular membranes to conduct some functions important to virus maintenance and replication [43,44]. So far, MCs have been isolated from human, animal, and insect viruses, and shown to contain many molecular chaperones for the corresponding host (Table 1). As cellular chaperones, HSPs function with respect to viral multiplication, replication, movement, assembly, disassembly, infection, uncoating, and virion morphogenesis in human, animal, and plant hosts (Figure 3).

### 4.1. HSPs Interact with Viral Proteins to Change the Activation Status

For multiplication in the host cells, a virus needs to produce an abundance of viral proteins. HSPs allow viral proteins to be folded into the activated form from the unfolded or misfolded form, converting viral proteins into soluble and activated forms [1,112]. The large T antigen (LT) protein of the Simian virus 40 (SV40) virus is multifunctional and can interact with HSP70 and HSP90, as well as the HSP p53 and DNA polymerase α. LT interacts with HSP90 to promote LT stabilization [95,113,114]. HSP90 promotes the activity and stability of the protease and helicase activities of nonstructural protein 3 (NS3) and the multifunctional protein NS5A of the hepatitis C virus (HCV). The HSP-inhibitory action of geldanamycin (GA) (Figure 4A), 17-allylamino-demethoxygeldanamycin (17-AAG), namely, tanespimycin (Figure 4B), or radicicol (Figure 4C) toward HSP90 reduce the NS3 protease activity required to liberate it from the NS2/3 protein precursor, and causes NS3 proteasomal degradation, allowing NS3 to be released from the NS2/3 protein precursor [55,56]. NACHT, LRR, and pyrin domain-containing protein 3 (NLRP3) was known as a key component of the inflammasome, and misfolded and degraded when HSP90, a client protein, was reduced. GA can inhibit HSP90 and reduce pyroptosis through the means of NLRP3 [57]. The protease activity of NS3 is displayed by HSP90 targeting the helicase domain of NS3, so the deletion of the helicase domain of NS3 can stop the degradation of NS3 triggered by the HSP90 inhibitor, and contribute to the stabilization of this truncated protein. NS5A of HCV can interact with HSP90, and facilitate the stabilization of this protein [58].

### 4.2. HSPs Are Involved in the Assembly of Macromolecular Protein Complexes

The tombusvirus replicase complex contains viral proteins of RNA-dependent RNA polymerase and p33 replication cofactor, as well as host proteins, such as HSP70. The process of assembly takes place on the peroxisomal membrane [115,116,117,118]. The replicase complex of Red clover necrotic mosaic virus (RCNMV) consists of virus-encoded proteins, such as p27, and host-encoded proteins, such as HSP70 and HSP90, etc. After the interaction between P27 and HSP was treated with the HSP inhibitors 2-phenylethynesulfonamide (PES), otherwise known as phenylacetylenylsulfonamide or pifithrin-µ (PFTµ) (Figure 4D) or GA, p27 alone was purified from the replicase complex without HSP70 and HSP90 via immunopurification. HSP70 and HSP90 participate in different routes in the assembly of the replicase complex [109].

### 4.3. HSPs Are Involved with the Subcellular Localization of Virus Components

HSPs can interact with the coat protein of the virus in the formation of the viral particle. After the interaction between HSP70 and the Tomato yellow leaf curl virus (TYLCV) coat protein, the subcellular localization of TYLCV changed to the nucleus from the cytoplasm, so this interaction promoted the formation of the viral particle with proliferation activity [119]. An in vitro assay involving RNA-dependent RNA polymerase (p92pol, RdRp), the p33 replication cofactor (the key protein in the recruitment of the viral RNA of Tomato bushy stunt tombusvirus, TBSV, in replication), and Ssa1p (a HSP70 homolog encoded by the yeast *S. cerevisiae*) indicated that Ssa1p helps p33 and RdRp insert into membranes [120]. A temperature-sensitive mutant, ssa1(ts), of the HSP70 family, which came from the yeast host, cannot help the viral replication proteins of TBSV to locate the membrane [121].

### 4.4. HSPs Are Involved in Viral Replication

HSPs play many roles in the replication of different viruses, including both DNA viruses and RNA viruses. For human and animal viruses, the reverse transcriptase (RT) of duck hepatitis B virus (DHBV), and human hepatitis B virus (HBV), which are members of the *Hepadnaviridae* family, mediate the incorporation of viral pregenomic RNA (pgRNA) into nucleocapsids and promote the reverse transcription of pgRNA into DNA [122]. Cooperating with HSP70 and HSP40, and the cochaperones Hop and p23, HSP90 can initiate and maintain the ability of the RT, and facilitate incorporation of the pgRNA into nucleocapsids [66,67,123]. Cdc37, a cochaperone that acts as an additional component of the IkappaB kinase complex, can interact with the RT of DHBV independently of HSP90, promoting transcription of pgRNA and assisting incorporation of pgRNA into nucleocapsids [68,70,71]. RdRp of influenza virus A consists of three subunits, PB1, PB2, and PA. In infected cells, HSP90 can interact with PB1 and PB2 of RdRp, and relocalizes to the nucleus, promoting viral RNA synthesis [86,87]. GA or 17-AAG (a GA analog), inhibitors of HSP90, can promote the degradation of PB1 and PB2 and decrease the amount of the polymerase complex. As a result, virus-derived RNAs were also reduced [89]. In an example of a human virus, once HSP90 was inhibited by GA, radicicol or small interfering RNA (siRNA), the activity of the large subunit of the vesicular stomatitis virus (VSV) polymerase was reduced and the replication of the virus was decreased (Figure 4) [86]. HSP90 can induce RdRp of herpes simplex virus type 1 (HSV-1) to form virus-induced chaperone-enriched (VICE) compartments, which can facilitate viral replication in the nucleus [124]. The disrupting action of GA toward HSP90 can influence the mislocation of VICE into the cytoplasm instead of the nucleus, with the action resulting in the degradation of RdRp, and a decrease in viral DNA replication [77,78].

For plant viruses, HSP70 can activate an N-terminally truncated RNA-dependent RNA polymerase (RdRp) of the tomato bushy stunt tombusvirus (TBSV) in vitro with the help of neutral phospholipid-containing phosphatidylethanolamine and phosphatidylcholine [125]. An in vitro cell-free system, consisting of TBSV replication protein, the viral RNA replicon, HSP70, rATP, rGTP, and a yeast cell-free extract, was assembled to form the protein complex, which can synthesize new TBSV replicon RNA [108]. A four-membered SSA subfamily consists of HSP70 genes in the yeast model. When ssa1 (ts) was used as a temperature-sensitive protein in the assay of the yeast model host, the viral replication proteins failed to localize onto the membrane instead of being distributed in the cytosol, so the viral replication complex lost replicase activity [121]. The above results indicated that HSP70 can promote virus replication via interaction between HSP70 and RdRp. In order to produce VRC containing Ssa1/2p protein (homolog of HSP70), *S. cerevisiae* was infected by cucumber necrosis virus (CNV) to become the functional host. Overexpressed assays of either Ssa1p or Ssa2p indicated that they both promote viral RNA replication [118]. For the Pepino mosaic virus (PepMV), the viral complex consists of PepMV capsid protein (CP) and HSP70, the latter promoting viral replication as verified experimentally by using virus-induced gene silencing (VIGS) against HSP70, or spraying the HSP inhibitor quercetin onto *N. benthamiana* plants [126]. In addition, HSP90 can promote the replication of the Bamboo mosaic virus (BaMV) by this protein interacting with RdRp and the untranslated region (UTR) of BaMV in *N. benthamiana* [127].

### 4.5. HSPs Are Involved in Viral Translation

HSP90 can promote the translation of protein A of the flock house virus (FHV), a member of the *Nodaviridae* family, but lacks polymerase activity [84]. The use of GA can reduce the levels of polysomes translating protein A, but cannot influence the translation of other viral proteins or cellular RNA related to polysomes [83]. HSP90 can help the Nsp3 protein of rotavirus to fold and stabilize [104]. Nsp3 relocalizes the cytoplasmic poly(A)-binding protein to the nucleus, shutting off cellular translation [128,129,130]. The action of HSP90 inhibitors, such as 17-AAG (Figure 4B) or alvespimycin (17-DMAG) (Figure 4E), can reduce Nsp3 translation, and disrupt nuclear translocation of the poly(A)-binding protein, thus suppressing viral replication [104,131].

### 4.6. HSPs Are Involved in Viral Infection

HSPs can regulate viral infection by modulating host processes and related cellular proteins via the model of HSP90 and HSP70, localized on the surface of some cells [132]. HSP90 and HSP70, localized on the cell surface, can internalize dengue virus (DENV) and infectious bursal disease virus (IBDV) [54,133]. During the course of DENV infection, HSP90 can interact with the viral receptor complex [133]. Some environmental factors, such as heat shock, can stimulate the upregulation of HSP90 and HSP70 localization onto the cell surface [134]. As a consequence, transient heat shock can enhance the infectivity of DENV [135]. Moreover, transient heat shock has been shown to increase cell surface expression of HSP90 and HSP70 and also increase dengue virus infectivity [108]. Applying anti-HSP90α or anti-HSP70 antibodies in monocytes and macrophages infected by DENV can reduce viral infection [54]. Both HSP90 and anti-HSP90 can also reduce IBDV infectivity [89].

### 4.7. HSPs Are Involved in Viral Movement

Viral movement in plant hosts is associated with intracellular movement, local movement between cells, and long-distance movement via the vascular system of the plant [136]. Different modes of movement are always involved with the interaction of HSP and virus-encoded proteins, such as movement protein, coat protein, RdRp, etc. [136,137,138,139]. HSP70 can interact with the movement protein of the Abutilon mosaic virus (AbMV); if HSP70 is silenced, the movement of the virus is restricted, but the replication of the virus is not affected [140]. When the movement protein interacts with host factors, including HSPs, to form the macromolecular complex, the movement protein of the virus facilitates its own long-distance movement by changing the structure of plasmodesmata in the plant cells [141,142,143,144]. HSP70 can interact with the CP of the TYLCV and form functional complexes with multiplication activity, which relocate from the cytoplasm to the nucleus. When HSP70 activity was inhibited by the HSP inhibitor quercetin, relocalization of the function complexes occurred from the nucleus to the cytoplasm. These results indicated that HSP70 contributes to the movement of the viral complexes within the plant cells [119]. Analysis of RdRp mutants of Brome mosaic virus (BMV) indicated that the BMV mutant was incapable of long-distance movement [137]. Closteroviruses consist of a long body and a short tail, which is composed of the major capsid protein (MCP) and the minor capsid protein (CPm), and HSP70h (a HSP70 homolog), which is encoded by the virus [50,145]. Driven by ATP binding to HSP70h, the virions were translocated via a plasmodesmata channel after the tail became attached to the plant cell wall [50]. The genome of the Beet yellows virus (BYV), also a member of the *Closteroviridae* family, can encode a HSP70h protein via open reading frame 3 (ORF3) [146]. HSP70h is involved with two modes of movement. The long-distance movement of the virus is associated with HSP70h and p20 [147]. HSP70h is also involved with intercellular translocation by encapsulating viral RNA [148], involving the inactivation of the HSP70h gene [149]. HSP70 was also involved with the BYV virion, binding to the plasmodesmata channel by the tail of BYV, to achieve translocation in the plasmodesmata channel, and to achieve disassembly and exposure of viral RNA [1].

### 4.8. HSPs Are Involved with the Assembly of Viral Particles

Viral capsids consist of one or more types of structural proteins, with each protein type containing over 1000 subunits. Viral capsids can encapsulate the viral genome into a virion. The subunits of the structural proteins need to be correctly folded and assembled by chaperone proteins [150]. Replication of poliovirus and rhinovirus is conducted with the help of HSP90, then the maturation of the viral capsid proteins is facilitated by HSP90 [93]. The capsid subunit of the picornavirus is digested into the activated state from the status of the P1 precursor protein and then assembled into the viral capsid. The process of cleavage of the precursor protein involves HSP90 and the cochaperone, p23. Nevertheless, inhibition of HSP90 does not affect viral protease activity, and the role of HSP90 is to facilitate recognition of the P1 precursor protein by the viral protease [93]. For hepadnaviruses, the duck hepatitis B virus was selected as an example, and HSP90 and p23, which is a chaperone partner for HSP90, were recruited into a ribonucleoprotein (RNP) complex driven by HSP70 and HSP40, providing energy, with the contribution of additional factors from the host. Then, the complex containing the viral polymerase, HSP90, p23, and additional factors can bind viral pregenome RNA after the conformational change, and trigger virion assembly [66]. For murine polyomavirus, a study of the model of A31 mouse fibroblasts, Sf9 insect cells, and the reticulocyte translation extract with the in vivo or in vitro assay indicated that HSP70 or its homolog can interact with viral capsid proteins, and facilitate assembly of capsid protein complexes to produce the virion, then transport it into the nucleus from the cytoplasm [151]. During capsid formation of the hepatitis B virus (HBV), Hsp70 and Hsp90 interact with core dimers to promote capsid assembly [152]. The application of HSP90 inhibitors, such as GA or novobiocin (Figure 4G), can decrease DNA replication of the vaccinia virus by downregulating the expression of the viral intermediate and late genes [153,154]. The interaction between HSP90 and the 4a core protein of the vaccinia virus facilitates the maturation and assembly of the viral capsid [98]. During vaccinia virus infection, inhibition of HSP90 by GA or novobiocin reduces viral DNA replication by specifically inhibiting intermediate and late, but not early, gene expression [103,104]. HSP90 interacts with the 4a core protein of the vaccinia virus, implicating HSP90 in the conformational maturation of the vaccinia capsid. In BYV in the *Closteroviridae* family, HSP70h can interact with a structural protein of BYV, and help to encapsulate viral RNA into the virion [148].

### 4.9. HSPs Are Involved with the Disassembly of Viral Particles

Maintenance of the stable viral particle within and outside the host facilitates virus survival and infectivity, and this stable structure also needs to be readily disassembled [150]. This disassembly process is assisted by HSPs. For instance, to generate the viral core particle with transcriptional activity, the outer capsid of the mammalian reovirus needs to be disassembled. This process involves the release of mu1 protein in the cytoplasm, which contains the central and delta fragments, with the help of HSP70. In vitro assays indicated that the degree of increase or decrease in the HSP70 level can increase or decrease the release of the delta fragment, respectively. Viral disassembly of the outer capsid was strengthened or weakened by being accompanied by increased or decreased levels of HSP70, respectively [155].

### 4.10. HSPs Are Involved in Viral Transport

In the influenza virus, the monomers of hemagglutinin are folded and assembled into trimeric structures with the aid of the 77 kDa molecular weight chaperone, and are translocated across the membrane of the rough endoplasmic reticulum. The mutant hemagglutinin could not be translocated because it was not properly folded by the chaperone protein [156]. In addition, in the human immunodeficiency virus type 1 (HIV1), the envelope glycoprotein gp160 was also translocated from the endoplasmic reticulum by binding between an HSP70 homolog and special polypeptides with a conserved sequence. The HSP70 homolog possesses the ability to recognize (through the conserved sequence) and fold the glycoprotein gp160 [157].

## 5. Viral Infection Can Change HSPs from Host and Virus

### 5.1. Human or Animal Viral Infection Can Change the Intracellular Distribution of HSPs from Mammal Cells

When African green monkey kidney cells (Vero CCl81) were infected by human HSV-1, the expression level of Hsc70, HSP70, and HSP40 did not differ between infected cells and uninfected cells; nevertheless, the distribution of these three proteins was found to change in the cells upon infection [124]. In common with the previous study, the amount of HSP70 from Hep2 cells did not change upon infection by the respiratory syncytial virus (RSV^a^), but the amount of HSP70 increased in the lipid-raft membranes [158].

### 5.2. Viral Infection Can Enhance the Amount of HSPs in Mammal Cells

In the model of mice infected by herpes simplex virus (HSV), a molecular chaperone with molecular weight of 57,000 Da, namely p57, accumulated in the cell nucleus and cytoplasm, similar to the response of mouse cells exposed to heat stress [159]. In HeLa cells infected with adenovirus type 5 (Ad5) and HSV-1, HSP70 expression was upregulated, according to run-on transcription assays. Furthermore, the induction of HSP70 has also been reported in certain host cells in response to infection by various viruses [30,160,161].

### 5.3. Viral Infection Can Enhance the Amount of HSPs in Plant Cells

Studies have indicated that a number of plant viruses can upregulate the level of HSPs in infected plant hosts. For instance, TYLCV, PepMV, AbMV, Turnip mosaic virus (TuMV), Turnip crinkle virus (TCV), Tobacco mosaic virus (TMV), Potato virus X (PVX), Cucumber mosaic virus (CMV), Watermelon mosaic virus (WMV), Rice stripe virus (RSV^b^), CNV, and RCNMV enhanced the levels of HSP70 or Hsc70 in tomato, *Arabidopsis thaliana*, tobacco, and rice [1,9,96,109,111,119,140,162,163]. Furthermore, to measure gene expression levels, members of the HSP70 gene family were analyzed using next-generation sequence technology with the model of *N. benthamiana* infected by CNV. Hsc70-5, Hsc70-1, and Hsc70-2 were strongly induced by CNV, with the ratio of plus-CNV vs. mock being more than 100-fold; Hsc70-2, Hsc70-15, HSP70-8, and HSP70-18 were moderately induced, with the ratio being between 4.5- and 100-fold; and HSP70-16 and HSP70-17 were mildly induced, with the ratio being between 1- and 4.5-fold [1]. RSV^b^ can also upregulate the mRNA level of HSP20 in tobacco and rice [110]. Infection of *N. benthamiana* by BaMV also upregulated NbHSP90 [127]. The above studies indicate that HSPs are essential for virus infectivity.

### 5.4. Viral Infection Can Enhance the Amount of HSPs Encoded by Itself

The *Closterovirus* family consists of BYV, Beet pseudo-yellows virus (BPYV), Citrus Tristeza virus, and Lettuce infectious yellows virus, etc. BYV can encode a HSP70 homolog, namely HSP70h, identified by sequence comparison with HSP70 from the plant [145,164]. After *N. benthamiana* plants were infected by BYV, HSP70h was primarily accumulated in cells in the phloem tissue, and secondly in close proximity to plasmodesmata and inside the plasmodesmata channels [148]. The BPYV genome can also encode HSP70 via sequence analysis and comparison [165].

### 5.5. The Mechanism of HSP Transcription Is Triggered by Viral Infection

In addition to the regulation of HSFs to HSPs, the transcription mechanism of HSPs may be involved with other pathways or factors following viral infection. Some proteins encoded by the polyomavirus, SV40, and adenovirus can promote transactivation of the viral genome [166,167,168,169,170,171,172,173,174]. Polyomavirus large T and SV40 large T antigens can bind to or recognize the special sequence in the region of the promoter of the viral genome [175,176]. The adenovirus ElA region can increase the abundance of human HSP70 mRNA of humans by approximately 100-fold [160,161]. After IC4 and 293 cell lines are induced by dexamethasone, expression of ElA alone can promote the transcription of human HSP70 [30]. Large T antigen of SV40 can transactivate monkey HSP70 [177]. The α (or immediate-early) group encoded by HSV can promote the transcription of specific deoxypyrimidine kinase (dPyK) mRNA by binding to the promoter of the corresponding gene [178]. A study found that the transcriptional activation of HSP70 induced by Ad5 and HSV-1 was transient and then turned into a repressed status, and there was no binding between HSF and HSP70 during this process [30]. This phenomenon indicated that the transcriptional activation of HSP70 is regulated by factors other than viral proteins, but not HSF. Another study indicated that the expression of thymidine kinase and HSP70 from HeLa cells infected by HSV-1 was simultaneously induced; the study also showed that a factor of the immediate-early group from the virus can regulate the expression of HSP70 [30]. The early regional group in the polyomavirus genome can encode large, middle, and small T antigens, and these proteins can interact with the promoter of the human HSP70 gene [177]. Compared with entire antigens containing the large, middle, and small T antigens, the large antigens of the polyomavirus early region play a partial role in transactivation [177]. Nevertheless, the assays of a deletion of the binding sequence of the large T antigens in the promoter of the HSP70 gene indicated that the large T antigens can still promote transcription of HSP70 mRNA, and this effectively means that the transcription effect of HSP70 existed as an indirect action, rather than simple protein–DNA binding [177]. Two sites, located −110 and −170 bp upstream of the human HSP70 promoter, were also particularly recognized by polyomavirus large T antigen, where the sequence carries the characteristic of diversion copies of “GPuGGc” [177,178,179,180]. The sequence features of these two sites are present as a pentanucleotide sequence of “GPuGGc” arranged in direct orientation, with an interval of 7 bp in site one and the same pentanucleotide sequence is arranged in an inverted orientation with an interval of 9 bp (Figure 2) [177]. Guanine N7-methylation in two sites inhibits the binding action of polyomavirus large T and SV40 large T antigens to a specific sequence [177]. Nevertheless, the transactivation to the HSE of HSP70 against HSF is not completely identical in the above two sites [177]. The “GATTGGCTC” sequence centered at −64 bp of the HSP70 promoter is responsible for the transactivation of HSF (Figure 2) [31].

## 6. HSP Inhibitors Can Downregulate HSPs at Many Levels

### 6.1. Quercetin Can Decrease the Expression Level of HSP70

Quercetin, with the chemical name 3, 3′, 4′, 5, 7-pentahydroxyflavone (Figure 5A), which belongs to the flavonoid family of compounds, is present in flowers, leaves, and fruits of many plants [181]. Quercetin is an important traditional medicine because of its antihypertensive, anticancer, anti-inflammation, antidiabetic, antioxidant, and antiviral activities, etc., as well as its low cytotoxicity. These activities explain why quercetin is widely used in the medical industry [181,182,183]. Quercetin can inhibit cytokine production, and reduce the activity of cyclooxygenase and lipoxygenase, then exerts powerful anti-inflammatory effects [184,185,186]. However, due to its poor water solubility, chemical stability, and absorption properties, quercetin’s bioavailability is usually less than 10% [187]. At present, the bioavailability of quercetin was enhanced using encapsulation technologies and crystal engineering [188,189,190].

Quercetin can downregulate the expression level of HSP70 in humans, animals, and plants [111,119,120,126]. A number of human examples confirm the downregulation of HSPs in response to quercetin treatment. Examples are as follows: Compared with unstressed normal cells, HSP70 was observed to play the role of chaperone in human cancer cells under stress [191]. After HeLa and COLO320 DM cells were treated with quercetin, the amounts of HSP90, Hsc70, HSP70, HSP47, and HSP28 were analyzed. There was only HSP70 downregulated at the level of mRNA. [36]. As with HeLa and COLO320 DM cells, U937, a human monoblastic leukemia cell line, was treated with quercetin at the low dosage of 10 µM for 30 min, in response to which, HSP70 was significantly downregulated [191]. Quercetin also decreased the expression of the low-molecular-weight HSP phospho (p)-HSP27 from the human tongue cancer cell line SCC25 at the protein level [192].

In addition, quercetin can also inhibit the expression of HSP70 in many plants. For instance, when *N. benthamiana* leaves were treated with quercetin at a concentration of 1 mM for 10 min, cytosolic NbHSP70 mRNA was quickly inhibited to 80% of the level in untreated *N. benthamiana* leaves using the Northern blot assay [120]. For *N. benthamiana* leaves infected by PepMV, the amount of HSP70 following quercetin treatment was reduced by 50–73% compared with that of the control treatment after 4 days of spraying, using the Western blot assay [126]. In addition, quercetin at a concentration of 400 µM also slightly reduced the amount of HSP70 in the cytoplasm in tomato leaves, but did not influence the amount in the nucleus [119]. HSP70 consists of a number of subfamilies, each of which contains many members. The expression of these members of HSP70 exhibits a tissue-specific expression pattern. The majority of HSP70 members are responsive to drought and heat stress [193]. These results indicated that the identity of the HSPs downregulated by quercetin may be a selective feature.

### 6.2. Quercetin Regulates HSF at Many Levels

Although quercetin can inhibit HSPs at the mRNA level, the mechanism has not been fully elucidated. Some studies have indicated that quercetin can regulate HSFs at many levels in different hosts. Quercetin can decrease the expression of Hsf2 in MDA-MB-231, a human breast cancer cell line [194]. Quercetin can also decrease the basal level of HSF1 in HeLa cells [34,194]. In addition, the P-gp gene, as a multidrug resistance (MDR) gene, which contains HSE promoter, was found to downregulate after being treated with quercetin via downregulating HSF. This inhibition activity against the P-gp gene presents a dose-dependent manner in FM3A/M and P388/M MDR cells (Figure 6) [195].

Data from HeLa cells indicated that heat shock increased the concentration of a phosphorylated form of HSF1, and that quercetin could significantly decrease the amount of the phosphorylated form of HSF1 following heat shock treatment, or in the untreated control. Meanwhile, treatment with quercetin did not change the amount of the unphosphorylated form of HSF1, regardless of treatment conditions [34]. In addition, another study indicated that the condition factors of elevated temperature, low pH, urea, detergents, or Ca^2+^ can activate HSF levels in cytoplasmic extracts of HeLa cells untreated with heat shock [32,196]. However, the HSF activated by heat shock is inhibited by adding quercetin [197]. Meanwhile, the activated HSFs from cytoplasmic extracts following exposure of the cells to heat shock can bind HSEs during the binding reaction of added quercetin. The above results indicated that quercetin can inhibit the status of the activated HSF, but did not influence the binding mode between HSF and HSE (Figure 6) [197].

Finally, quercetin can selectively inhibit the binding of HSFs and HSEs [194]. Some studies indicated that quercetin treatment prior to heat shock could inhibit the binding of HSF and HSE to HSP27 and HSP70 in HeLa cells, but was defective in action for MDA-MB-231 cells (Figure 6) [194]. In addition, quercetin could inhibit the binding of HSF to HSE oligonucleotides during heat shock treatment in COLO 320DM cells in vivo [197].

### 6.3. Flavonoid Compounds with Structural Similarity to Quercetin Can Decrease the Expression Level of HSPs or Change the Conformation of HSPs

Expression of HSP110, HSP90, HSP70, HSP72, HSP47, and HSP40 from COLO320 DM cells was upregulated in response to heat stress. Flavone, kaempferol, and genistein can inhibit the expression of these HSPs (Figure 7A–C). Rutin can inhibit the expression of many HSPs. For induction of HSP90, the strongest effect was triggered by kaempferol or quercetin, then flavone or genistein (Figure 7D) [36]. When MCF-7, a human breast cancer cell line, was treated with EGCG at 100 µM, HSP70 and HSP90 levels were significantly decreased at the level of mRNA and protein. On the other hand, the levels of HSP110, HSP60, HSP40, and HSP27 were almost unchanged following exposure to EGCG. Similar to the results of the previous study, when the promoters of HSP70 or HSP90 were fused to the luciferase gene (*LUC*), the luciferase activity was also decreased by EGCG [198]. In addition, EGCG can bind to the C-terminal region of HSP90, which implies an ATP binding site, so as to change its conformation to prevent the HSP90 from achieving dimerization [199].

### 6.4. Flavonoids with Structure Similar to Quercetin Regulate HSFs

COLO 320DM cells were treated at 43 °C for 90 min and then treated with genistein or flavone at 100 or 150 µM, respectively. The electrophoretic mobility shift assay indicated that genistein cannot disturb the binding of HSFs to HSEs, whereas flavone partly disturbs the binding of HSFs to HSEs [197].

### 6.5. KNK437 Can Decrease the mRNA Level of HSPs

A novel benzylidene lactam compound, 3-(1,3-benzodioxol-5-ylmethylidene)-2-oxopyrrolidine-1-carbaldehyde (KNK437), namely, N-formyl-3,4-methylenedioxy-benzylidene-*γ*-butyrolactam (Figure 4F), can strongly inhibit the expression of HSPs at the mRNA level in a dose-dependent manner in COLO 320DM, HeLa S3, and SCC VII cells. For instance, KNK437 can inhibit the synthesis of HSP70, HSP40, and HSP105 in COLO 320DM cells [200]. The inhibitory activity toward HSPs (including HSP110/HSP105, HSP72, and HSP40) of KNK437 is significantly higher than that of quercetin. HSP70 mRNA was significantly induced after heat treatment at 42 °C for 90 min, and 100 µm KNK437 almost completely inhibited the accumulation of HSP70 mRNA after heat shock. [200]. KNK437 and quercetin can inhibit the mRNA level of HSP30, HSP47, and HSP70 in *Xenopus laevis* cultured cells, and the degree of inhibition achieved by KNK437 is also significantly greater than that of quercetin. Exposure of A6 cells to 100 µM KNK437 and quercetin significantly reduced the accumulation of HSP30 by 100% and 50%, respectively [201]. The inhibitory action of KNK437 toward HSPs can antagonize the induction of thermotolerance by sodium arsenite [200]. Nevertheless, the mechanism of inhibiting HSPs by KNK437 is different from that of quercetin. Some studies speculate that the action mechanism of KNK437 involves two processes, either inhibiting the activation of HSP through the process of phosphorylation, or inhibiting the binding of HSFs to the HSEs of HSP [200].

### 6.6. Inhibitors PES and GA Can Disrupt the Interaction between HSP and Its Client Protein

Here, human and animal viruses were used as case studies. The small molecule PES can inhibit binding between the HSP p53 and mitochondria (Figure 4D) [202]. PES can interact with HSP70, but not Hsc70, the glucose-regulating protein and molecular chaperone GRP78, and HSP90 in mammalian cell lines, as well as interacting with DnaK, the bacterial ortholog of mammalian HSP70. Meanwhile, deletion analysis assays indicated that the binding sites of PES to HSP70 associate the carboxyl (C)-terminal domain with the function of substrate-binding, but not the amino (N)-terminal domain with the activity of ATPase for human HSP70 [203].

GA is a molecule with antimicrobial activity and antitumor activity, which has been isolated from *Streptomyces hygroscopicus* var. *geldanus* var. *nova* [204]. GA and its analogues 17-AAG and 17-DMAG can target N-terminal ATP binding site of HSP90, then inhibit the proliferation. The therapeutic application of geldanamycin has been limited due to its poor water solubility and severe hepatotoxicity [205]. GA can bind to the N-terminal ATP/ADP-binding domain of HSP90, which possesses a pocket-shaped binding site with ATP, so that GA can disrupt the interaction between HSP90 and its client protein by competing with ATP [206]. Replication of the Chikungunya virus (CHIKV) is dependent on HSP90 via the interaction of nonstructural P3 (viral RNA synthesis) or nonstructural P4 proteins (RdRp) with HSP90 in HEK-293T cells of human embryonic kidney cells. The application of GA can inhibit the replication of CHIKV and reduce the inflammation of mice infected with CHIKV [207].

Plant viruses were also selected as further case studies. P27, a viral protein from RCNMV, can promote virus replication via the model of the replication complex, which was recruited by HSP70 and HSP90. The immunopurification assay indicated that PES and GA can block the interaction of P27 with either HSP70 or HSP90, respectively. PES and GA can disrupt the assembly of the replicase complex and block negative-strand RNA synthesis of the virus [109].

### 6.7. Sodium Salicylate Changes the Expression of HSPs by Regulating HSFs

Treatments with heat shock and sodium salicylate can induce the expression and promote the phosphorylation of the same isoform of HSF (namely HSF1) in HeLa S3 cells, and these results indicate that these two treatments can promote HSP synthesis (Figure 7E). Cycloheximide, which is used as an inhibitor of protein synthesis, can block the binding activity of the HSF–DNA interaction induced by sodium salicylate, and fail against the treatment of heat shock. These results further show that the steps of HSF synthesis are triggered by two different treatments [37,208]. First, the amino acids in HSF1, which are phosphorylated, can differ, with heat shock treatment phosphorylating serine, whereas sodium salicylate phosphorylates threonine. Second, the degree of phosphorylation induced by sodium salicylate is significantly weaker than that achieved by heat shock, so the degree of DNA–HSF1 binding induced by sodium salicylate is significantly lower than that achieved by heat shock [208]. In addition, when HeLa S3 cells are exposed to two conditions, namely sodium salicylate or heat shock at 42 °C, HSP70 gene expression in the nucleus was not detected following sodium salicylate treatment, whereas it was significantly enhanced in response to heat shock [208]. These results indicated that HSP synthesis triggered by heat shock is stronger than that induced by sodium salicylate as a result of different action mechanisms.

### 6.8. Some Potential Inhibitors of HSP90

The derivatives of benzo[d]thiazole and 4,5,6,7-tetrahydrobenzo[d]thiazole displayed antiviral activity against influenza A (H1N1, H3N2) and influenza B viruses, and showed higher binding affinities for HSP90 [209]. Deguelin (Figure 4H), a naturally occurring rotenoid, can downregulate clients of HSP90 (Cdk4, Akt, eNOS, and MEK1/2) [210]. Deguelin bound to ATP and did not affect the ATP pocket in HSP90N [211]. These results indicated that deguelin might bind to the ATP pocket in HSP90C. Derrubone (Figure 4I) and silybin (Figure 4J) can target Hsp90C and then possess antitumor activity [212,213]. Virtual screening indicated that ginkgetin (Figure 4K) and theaflavin (Figure 4L) developed favorable, as well as crucial, interactions with the HSP90 ligand-binding pocket. Molecular dynamics simulations of these two natural molecules exhibited minimal fluctuations in the binding pattern of ginkgetin and theaflavin to HSP90, which retained crucial contacts throughout the simulation time. Ginkgetin and theaflavin were known as potent HSP90 inhibitors [214].

### 6.9. Dihydromyricetin-Based HSP70 Inhibitors

Dihydromyricetin, which is called ampelopsin, is extracted from *Ampelopsis grossedentata* leaf. Dihydromyricetin can target the ATP binding site of a 78 kDa glucose-regulated protein (GRP78) (heat shock 70 kDa protein 5) using the assays of molecular docking and surface plasmon resonance (Figure 7F) [215].

### 6.10. Rhodacyanine-Based HSP70 Inhibitors

A new series of rhodacyanine-based HSP70 inhibitors was developed, in which the cationic pyridin-1-ium or thiazol-3-ium ring of existing HSP70 inhibitors was replaced by a corresponding benzo-fused N-heterocycle (Figure 7G). These derivatives possessed antitumor activities, in part, by targeting HSP70. These putative inhibitors displayed differential antiproliferative efficacy against breast cancer cells (IC_50_ as low as 0.25 µM) versus nontumorigenic MCF-10A breast epithelial cells (IC_50_ ≥ 5 µM) [216].

### 6.11. Chromone Analogs Possessing Thiiran-2-Ylmethoxy or Oxyran-2-Ylmethoxy Substituents-Based sHsp Inhibitors

Chromone analogs possessing thiiran-2-ylmethoxy or oxyran-2-ylmethoxy substituents can induce significant abnormal HSP27 dimer formation in NCI-H460, a human lung cancer cell line (Figure 7H). The compounds can effectively produce abnormal HSP27 cross-linking, and then remarkably enhanced the antiproliferative activity of 17-AAG. Chromone analogs might have potential anticancer activity, which can modulate abnormal HSP27 dimerization as sHSP inhibitors [217].

## 7. The Antiviral Activity of HSP Inhibitors Indicated That These Compounds Can Regulate Viral Multiplication

### 7.1. Quercetin Possesses Potent Antiviral Activity

Based on HSP involvement in the process of viral uncoating, assembly of macromolecules, replication, assembly, movement, subcellular localization, transport of virus, and viral coating, we speculated that quercetin and its analogs could show antiviral activity through these processes. Quercetin possesses the ability to inhibit the replication of human viruses, such as HSV-1, poliovirus type 1, parainfluenza virus type 3 (Pf-3), RSV^a^, HBV, HCV, and influenza virus [218,219,220,221,222,223]. Madin–Darby canine kidney (MDCK) cells were incubated with three influenza virus A strains, then treated with quercetin; the results showed that the IC_50_ values of quercetin were 7.76, 6.23, and 2.74 µg/mL against A/Puerto Rico/8/34 (H1N1), A/FM-1/47/1 (H1N1), and A/Aichi/2/68 (H3N2), respectively [221]. For HepG2.2.15 and HuH-7 cell models, quercetin treatment also decreased the amount of hepatitis B surface antigen (HBsAg) and hepatitis B e-antigen (HBeAg) [219,220].

For animal viruses, quercetin can also inhibit the replication of Porcine reproductive and respiratory syndrome virus (PRRSV) in MARC-145 cells and Porcine alveolar macrophages, as well as Equid herpesvirus 1 in Vero cells [222,223,224]. The replication cycle of Equid herpesvirus 1 was controlled in the period 0–1 h after treatment with quercetin [222].

In addition, quercetin can also inhibit the activity of plant viruses, for instance, TBSV, TMV, TCV, and PepMV [120,126].

### 7.2. Quercetin Derivatives Exhibit Marked Antiviral Activity

For quercetin derivatives, many medical studies have indicated that these compounds can also possess antiviral activity. For instance, isoquercetin can inhibit influenza virus A (Oh7), with an ED_50_ value of 1.2 µM, which is significantly lower than that of quercetin (ED_50_: 48.0 µM) (Figure 5B) [225]. Isoquercetin can also inhibit Porcine H1N1 strains, with an ED_50_ value of 1.2 µM, the value of which was lower than that of quercetin (ED_50_: 48.2 µM) [226]. The hydroxyl groups on C3, C3′, and C5 of quercetin were substituted using phenolic ester, alkoxy, and aminoalkyl groups to generate a series of quercetin derivatives. Of the many quercetin derivatives, quercetin-3-gallate exhibits particularly high antiviral activity against porcine H1N1 strains, with an ED_50_ value of 9.1 µM, which was intermediate between those of quercetin and isoquercetin (Figure 5C) [226]. The hydroxyl group of quercetin was substituted using bromine, and the resulting quercetin-6, 8-dibromide had high antiviral activity against H1N1/pdm09, with the ED_50_ value being 6.0 µg/mL [227]. Quercetin-3-rhamnoside can inhibit the replication of HSV-1 and pseudorabies virus proliferating in HEp-2 cells and chick embryo fibroblast (CEF) cells, respectively (Figure 5D) [223].

### 7.3. Quercetin Analogs Exhibit Prominent Antiviral Activity

With a molecular structure similar to that of quercetin, hesperetin can also inhibit the replication of herpes simplex virus type 1, poliovirus type 1, parainfluenza virus, and RSV^a^ (Figure 5E) [218]. Isorhamnetin has a molecular structure similar to quercetin, but with a methoxy group on C3′ (Figure 5F). Compared with quercetin, in vitro and in vivo bioassays indicated that isorhamnetin has a higher antiviral activity than quercetin against influenza virus A/PR/08/34 (H1N1) [228]. In summary, hydroxyl groups on the sites at C3′ or C3 were associated with antiviral activity, and a methoxy group substituted by a hydroxyl group at the C3′ site can increase antiviral activity. Meanwhile, a glucoside moiety introduced at the C3 site can also significantly increase antiviral activity.

In addition, through the model of MDCK cells infected by influenza virus subtypes including A/H1N1, A/H3N2, and B virus, (−)-epigallocatechin-3-gallate (EGCG) displayed strong antiviral activity, with the EC_50_ values against the three subtypes being 28.4, 22.8, and 26.1 µM, respectively (Figure 5G) [229]. Apigenin is an active compound from flowers of *Paulownia tomentosa* trees, with a molecular structure similar to quercetin (Figure 7I). A bioactivity assay indicated that apigenin has inhibitory activity against enterovirus 71 (EV71), with the EC_50_ value being 11.0 mM. Nevertheless, two compounds, naringenin, and quercetin, exhibited no antiviral activity [230]. The replication of EV71 and sindbis virus (SV) is completed by the interaction of the 5’ UTR of viral RNA and a host factor, namely, the heterogeneous nuclear ribonucleoprotein A1 (hnRNP A1). As a consequence, hnRNP A1 is seen as a trans-acting factor of the EV71 genome [231,232]. Apigenin can disrupt the interaction between hnRNPs and EV71 genome RNA to block the replication of the virus and display inhibitory activity against EV71 (Figure 7I). The results indicated that the mechanism of antiviral activity of apigenin was different from that of quercetin and its analogs, and the EC_50_ value of apigenin blocking EV71 infection was 11.0 µM [230].

## 8. Future Directions

Viral diseases of humans, animals, and plants are difficult to cure, and traditional antiviral targets mainly involve RdRp and replicase, etc. These kinds of antiviral agents are prone to generate drug resistance. HSPs and other cellular chaperone proteins help viral proteins fold or assemble, assisting viruses to complete many significant life processes. At present, the action mechanism of the interactions between HSPs and HSP inhibitors, such as quercetin, GA, and their analogs, has been elucidated, and more details of the action mode of antiviral for HSP inhibitors have been investigated (Figure 8). So far, due to their low cytotoxicity and the relatively high level of tolerance to these chemicals shown by the hosts, these antiviral agents show good prospects for the application as antivirals, on humans, animals, or plants. In addition, it is highly significant that HSP inhibitors are also used as probes with which to study the interaction mechanism between virus and host. The current review is mainly focused on the mechanism of interaction between HSP70, HSP90, and the virus, although some questions remain regarding (1) the relationship between HSP70/HSP90 and the virus, (2) how HSP70 interacted with viral component transfer to HSP90, and (3) the role of other cochaperone or cellular proteins during the interaction between HSPs and viral components, with each question being worthy of further study.

## Figures and Tables

**Figure 1 genes-14-00792-f001:**
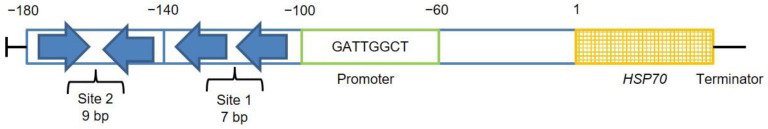
The structure of *HSP* genes. Abbreviations: HSP, heat shock protein.

**Figure 2 genes-14-00792-f002:**
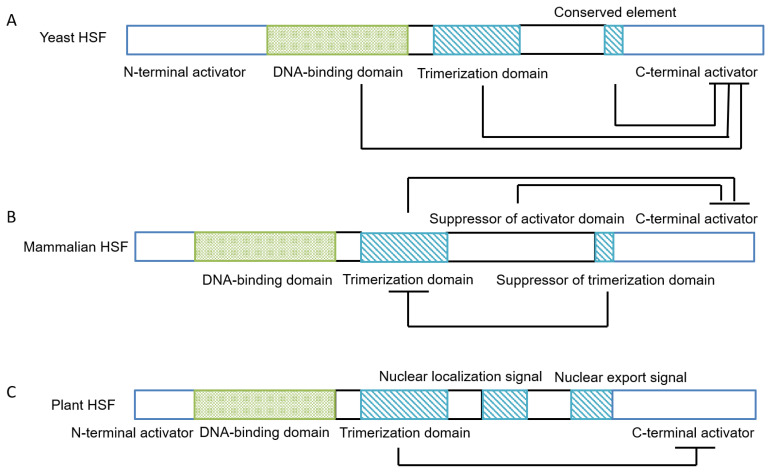
The structures of *HSF* genes. (**A**) Yeast. (**B**) Mammalian. (**C**) Plant. Abbreviations: HSF, heat shock factor.

**Figure 3 genes-14-00792-f003:**
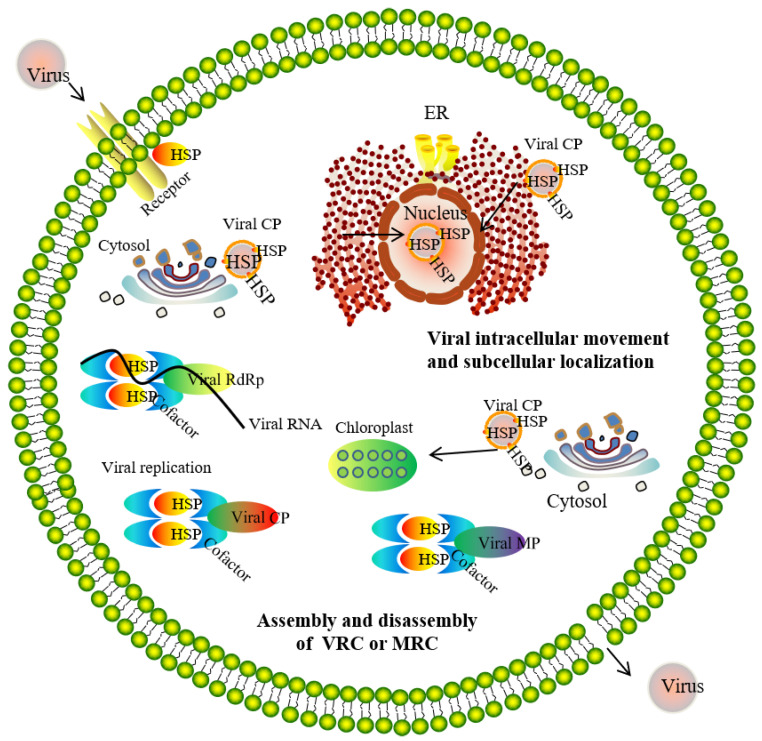
HSPs exhibit different functions in the viral macromolecular complex. Abbreviations: HSP, heat shock protein; ER, endoplasmic reticulum; MRC, macromolecular protein complexes; Viral CP, viral coat protein; Viral MP, viral movement protein; Viral RdRp, viral RNA-dependent RNA polymerase; VRC, viral replicase complex.

**Figure 4 genes-14-00792-f004:**
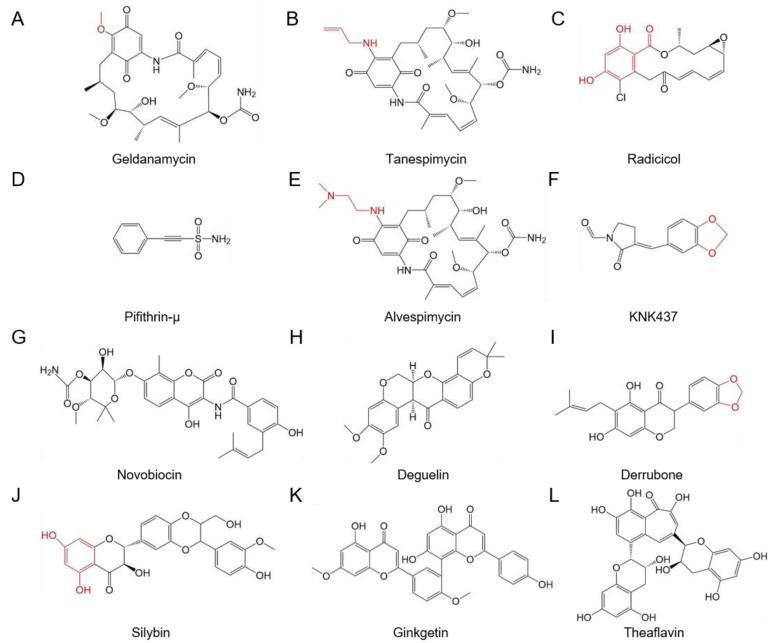
The molecule structure of HSP90 inhibitors. (**A**) Geldanamycin. (**B**) Tanespimycin. (**C**) Radicicol. (**D**) Pifithrin-µ. (**E**) Alvespimycin. (**F**) 3-(1,3-benzodioxol-5-ylmethylidene)-2-oxopyrrolidine-1-carbaldehyde. (**G**) Novobiocin. (**H**) Deguelin. (**I**) Derrubone. (**J**) Silybin. (**K**) Ginkgetin. (**L**) Theaflavin.

**Figure 5 genes-14-00792-f005:**
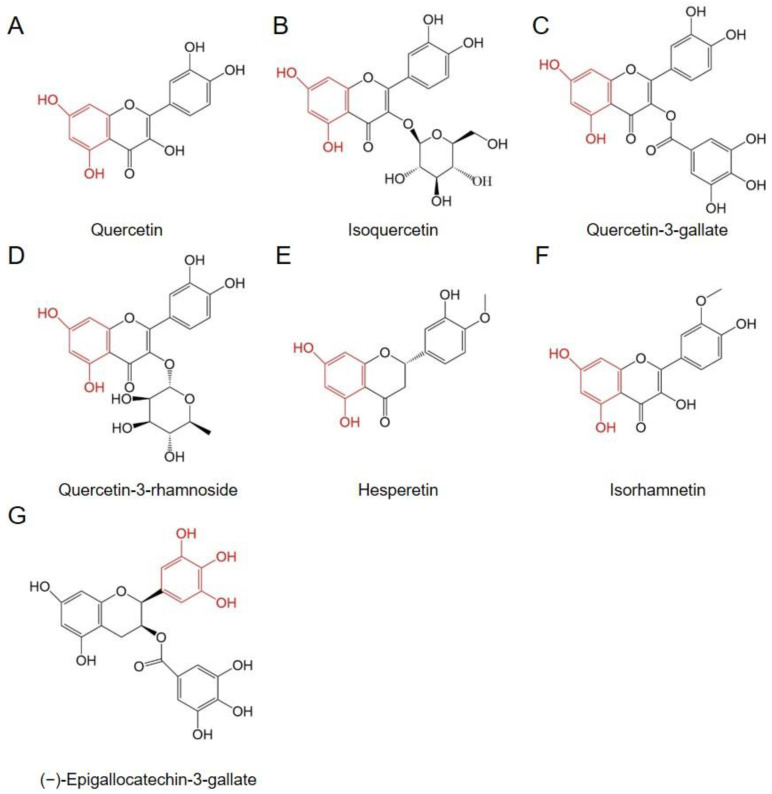
The molecular structure of quercetin and its analogs as HSP inhibitors. (**A**) Quercetin. (**B**) Isoquercetin. (**C**) Quercetin-3-gallate. (**D**) Quercetin-3-rhamnoside. (**E**) Hesperetin. (**F**) Isorhamnetin. (**G**) (−)-Epigallocatechin-3-gallate.

**Figure 6 genes-14-00792-f006:**
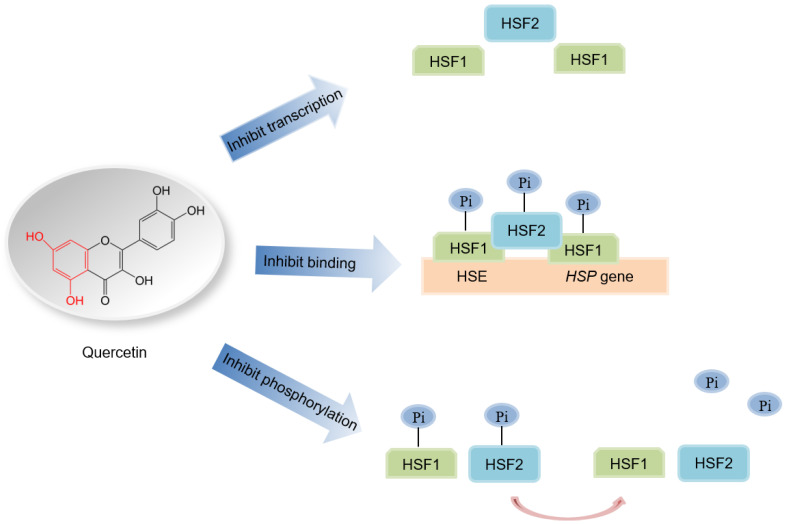
The action mechanism of HSP regulation by quercetin. Abbreviations: HSP, heat shock protein; HSF, heat shock factor; HSE, heat shock element.

**Figure 7 genes-14-00792-f007:**
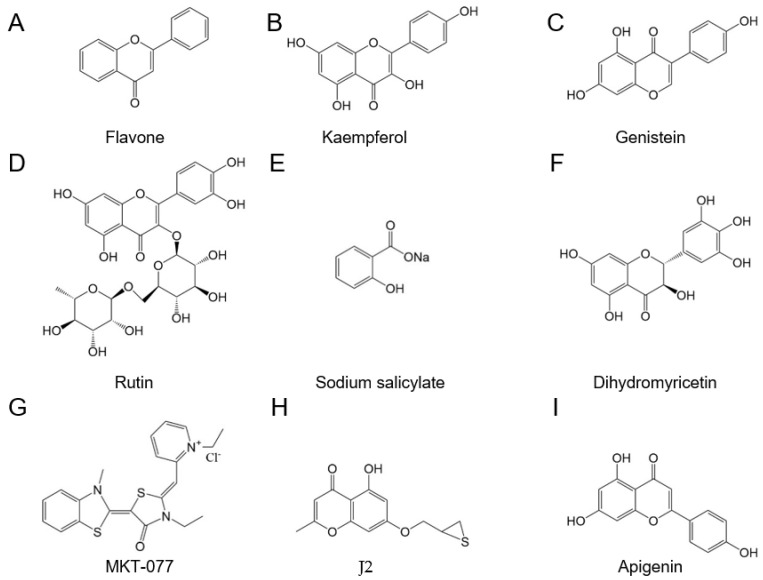
The molecular structure of flavonoids or other HSP inhibitors with the activity of HSP inhibition. (**A**) Flavone. (**B**) Kaempferol. (**C**) Genistein. (**D**) Rutin. (**E**) Sodium salicylate. (**F**) Dihydromyricetin. (**G**) Rhodacyanine-based (MKT-077). (**H**) Chromone analogs possessing thiiran-2-ylmethoxy or oxyran-2-ylmethoxy substituents. (**I**) Apigenin.

**Figure 8 genes-14-00792-f008:**
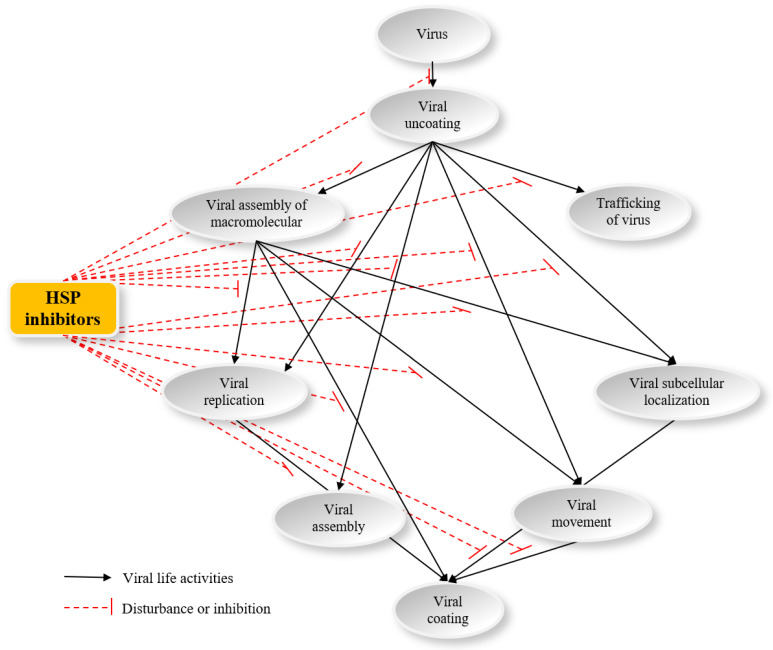
The antiviral mechanism of HSP inhibitors. Abbreviations: HSP, heat shock protein.

**Table 1 genes-14-00792-t001:** The interaction between the MC and viral components.

Viral Family	Virus	Viral Protein	Molecular Chaperones and Other Host Factors	Action and Process	Source of MC	References
*Arenaviridae*	LCMV	Nucleoprotein	HSP90	Antigen cross presentation	Human	[45]
*Baculoviridae*	BmNPV	Undetermined	HSP70	Viral replication	Animal	[46]
*Birnaviridae*	IBDV	VP2	HSP90	Virus Internalization	Animal	[47]
*Bunyaviridae*	LACV	L protein	HSP90	Protein stabilization	Human	[48]
*Closteroviridae*	BYV	Viral HSP70h	Microtubules protein and other proteins	Viral movement and assembly	Virus	[49,50]
*Coronaviridae*	IBV	Spike protein of IBV	HSPA8	Viral entry	Animal	[51]
	PEDV	membrane protein	HSP70	Viral replication	Animal	[52]
*Filoviridae*	Ebola virus	RdRp	HSP90	Virus propagation	Human	[53]
*Flaviviridae*	DENV	Viral receptor	Bip	Virus Internalization	Human	[54]
	HCV	NS3 Protease	HSP90	Cleavage at NS2/3 junction, NS3 function	Human	[55,56,57]
	HCV	NS5A	FKBP8, hB-ind1, HSP90, and HSP72	Replication complex formation/genome replication	Human	[58,59,60,61,62]
	JEV	NS3 and NS5	HSP70	Virus replication	Human	[63]
	TMUV	Capsid	HSP70	Viral replication, assembly, and release	Animal	[64]
	ZIKV	Capsid and viral RNA	HSP70	Viral entry, viral RNA production, and virion release	Humam	[65]
*Hepadnaviridae*	DHBV	P protein	p23, Cdc37, HSP40, and HSC70	Reverse transcriptase priming	Human	[66,67,68,69]
	HBV	P protein	Hsp70, Hsp40, Hop, and P23	Reverse transcriptase priming	Human	[66,70,71]
*Hepeviridae*	HEV	Capsid	HSP90	Intracellular transfer	Human	[72]
*Herpesviridae*	EBV	EBNA	HSP90	Cell proliferation	Human	[73]
	EBV	KH	PI90K	Apoptosis prevention	Human	[74,75]
	HCMV	MIE	PI3K	Expression of immediate early protein IE2	Human	[76]
	HSV 1	UL30	HSP70 and BAG3	Polymerase localization	Human	[77,78,79]
	HSV 2	UL30	HSP70 and BAG3	Polymerase localization	Human	[77,78,79]
	KSHV	K1	HSP40	Apoptosis prevention	Human	[80]
	KSHV	v-FLIP	IKK and Cdc37	Apoptosis prevention	Human	[81]
	VZV	ORF29F	HSP70 and BAG3	Localization of ORF29	Human	[82]
*Nodaviridae*	FHV	Protein A	HSP90	Replication complex formation	Animal	[83,84,85]
*Orthomyxoviridae*	Influenza A virus	PB1 and PB2	HSP90	Nuclear localization	Human	[86]
	Influenza A virus	RdRp	HSP90	Assembly and nuclear transport of viral RNA polymerase subunits	Human	[87]
	Influenza A virus	RdRp	HSP90	Viral RNA polymerase complex formation	Human	[87]
	Influenza A virus	RdRp	HSP90 and DnaJA1	RNA synthesis	Human	[88,89]
*Paramyxoviridae*	HPIV 2/3	L protein	HSP90	Protein stabilization and replication	Human	[48]
	MeV	MV-N	HSP70	Enhanced oncolytic activity	Human	[90]
	MuV	Viral polymerase	HSP90	Viral replication	Human	[91]
	SeV	-	TBK1, IRF3, and HSP90	Innate immunity activation and signal transduction	Human and animal	[92]
	SV 5/41	L protein	HSP90		Human and animal	[48]
*Picornaviridae*	Coxsackievirus	P1 capsid protein	p23	Cleavage of P1 into VP1, VP2, and VP3	Human	[93]
	Enterovirus 71 (EV71)	Viral particle	HSP70	Initial binding of virus to host cells	Human	[94]
	Poliovirus	P1 capsid protein	p23	Cleavage of P1 into VP1, VP2, and VP3	Human	[93]
	Rhinovirus	P1 capsid protein	p23	Cleavage of P1 into VP1, VP2, and VP3	Human	[93]
*Polyomaviridae*	SV 40	LT	HSP90	Stabilization of LT protein	Human and animal	[95]
*Potyviridae*	TuMV	RdRp	HSC70-3 and PABP	Viral replication	Plant	[96]
	WYMV	Undetermined	HSP23.6	Viral replication	Plant	[97]
*Poxviridae*	VACV	4a core protein	HSP70	Capsid assembly/virus gene expression	Human and animal	[98,99]
*Retrovirus*	HIV 1	tat	HSP90	Transcription/cell survival	Human	[100,101]
	HTLV	tat	HSP90	Transcription	Human	[101,102]
*Reoviridae*	Reovirus	σ1	HSP70 and p23	C′ trimerization of σ1	Human	[103]
	Rotavirus	NSP 3	HSP90	Dimerization of NSP3	Human	[104]
*Rhabdoviridae*	VSV	L protein	HSP90	Protein stabilization and replication	Human	[48]
	VSV	L protein and P protein	HSP60	Synthesizes capped mRNAs and initiates transcription at the first gene (*N*) start site	Human	[105]
*Togaviridae*	RUBV	p150 protein	HSP90	Replication of the viral genome	Human	[106]
*Tombusviridae*	TBSV	Viral replicase	CNS1p cochaperone of HSP70 and HSP90	Inhibited the assembly of the VRC and Viral RNA synthesis	Plant	[107]
	TBSV	Viral replicase	HSP70	Viral replication	Plant	[108]
	RCNMV	Viral replicase	HSP70 and HSP90	Promoting virus replication	Plant	[109]
	RCNMV	p27	HSP70 and HSP90	Interaction with HSP70 or HSP90	Plant	[109]
Uncertain	RSV^b^	RdRp	HSP20	Subcellular localization	Plant	[110]
	RSV^b^	RdRp	HSP70	Viral replication	Plant	[111]

Abbreviations: BAG3, Bcl-2-associated athanogene 3; Bip, binding immunoglobulin heavy-chain protein; BmNPV, bombyx mori nucleopolyhedrovirus; BYV, beet yellows virus; Cdc37, a cochaperone as additional components of the IKK complex; CNS, cyclophilin seven suppressor; DENV, dengue virus; DHBV, duck hepatitis B virus; DnaJA1, a member of the type I DnaJ/Hsp40 family; EBNA, EBV nuclear antigen; EBV, Epstein-Bar virus; EV71, enterovirus 71; FHV, flock house virus; FKBP8, FK506-binding protein 8; hB-ind1, human butyrate-induced transcript 1; HBV, Hepatitis B virus; HCMV, human cytomegalovirus; HCV, hepatitis C virus; HEV, hepatitis E virus; HIV1, human immunodeficiency virus 1; HPIV 2/3, human parainfluenza virus 2/3; HSP, heat shock protein; HSV1, herpes simplex virus type 1; HTLV, human adult T cell leukemia virus; IBDV, infectious bursal disease virus; IBV, infectious bronchitis virus; IE, immediate early; IKK, ikappaB kinase; IRF3, interferon regulatory factor 3; JEV, Japanese encephalitis virus; K1, a viral glycoprotein encoded by ORF1 of KSHV; KSHV, Kaposi sarcoma-associated herpesvirus; LACV, La Crosse virus; LCMV, lymphocytic choriomeningitis virus; L protein, the large subunit of the VSV polymerase; LT, large T antigen; MC, molecular chaperones; MeV, measles virus; NS3 Protease, nonstructural protein 3; NS5A, nonstructural protein 5A; NSP, nonstructural protein; ORF29F, open reading frames 29F; p27, a 27 kDa auxiliary protein; PABP, poly(A)-binding host proteins; PB1 and PB2, polymerase basic protein 1 and 2; PEDV, porcine epidemic diarrhoea virus; PI3-K, phosphatidylinositol 3-kinase; RCNMV, red clover necrotic mosaic virus; RdRp, RNA-dependent RNA polymerase; RSV, rice stripe virus; RUBV, rubella virus; SeV, Sendai virus; SV 5/41, simian virus 5/41; Tat, transcriptional activation via an RNA target; TBK1, TANK binding kinase I; TBSV, tomato bushy stunt virus; TMUV, Tembusu virus; TuMV, turnip Mosaic Virus; UL30, the viral polymerase; VACV, vaccinia virus; v-FLIP, viral FLICE-inhibitory proteins; VP, capsid protein; VSV, vesicular stomatitis virus; VZV, varicella zoster virus; WYMV, wheat yellow mosaic virus; ZIKV, zika virus.

## Data Availability

The data that support this study are available in the article.

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
