# Peer review of "Mode of Action of Heat Shock Protein (HSP) Inhibitors against Viruses through Host HSP and Virus Interactions"

_genes, 2023, doi:10.3390/genes14040792_

Round 1

Reviewer 1 Report

This manuscript is a compilation of interactions of HSP inhibitors and HSPs, HSFs and HSP genes.  Authors may further highlight research directions and luminescence the path for productive research. In the introduction section, perhaps, the scope of this review should be stated.  How were the citations selected?  You may describe what keywords were used for the literature search, which databases, and what years covered.  How were the HSP inhibitors selected for discussion?  Some chemicals that directly interact with HSP are not included, which, for example, dihydromyricetin interacts with HSP78 (GRP78) (https://doi.org/10.1093/jn/nxab057).

You may explicitly state that the HSPs you mean in this review are those in the hosts or the viruses. 

L442: Section 6 is the core of this review.  Authors may clarify that what HSP inhibitors have been specifically identified their direct interaction with HSPs (or HSFs or HSEs) in vitro and in vivo and what methods were used. 

Authors in this section spent large pagination to describe quercetin and its analogs.  It is well known that flavonoids such as quercetin have low bioavailability. Authors may want to elaborate their bioavailability.

Minor comments. 

L2: Tittle: Perhaps the title can be revised to “Mode of Action of Heat Shock Protein (HSP) Inhibitors Against Viruses”.  “Through HSP/Virus Interaction” does not seem to add additional meaning. Or you may have the title as: “Mode of Action of Heat Shock Protein (HSP) Inhibitors Against Viruses through host HSP and Virus Interactions.”

L11-12: You may want to revise this sentence. This sentence may be revised to: Molecular chaperones mediate the correct refolding of mis-folded proteins after stress-induced denaturation to regain their functions. OR Mis-folded proteins after stress-induced denaturation can regain their functions through correct re-folding with the aid of molecular chaperones.

L18-19: … their potential use as antiviral agents

L35: delete “, etc.”

L45: delete “also”

L69” sHSPs possess…

L250 and Figure 3: chemical names can be labelled under each chemical structure.

L362, Figure 4: define acronym in Figure 4 in the figure legend.

L507, Figure 6: are those direct interactions between quercetin and HSF, HSE and HSP gene?

Author Response

Reviewer 1:

This manuscript is a compilation of interactions of HSP inhibitors and HSPs, HSFs, and HSP genes. Authors may further highlight research directions and luminescence the path for productive research. In the introduction section, perhaps, the scope of this review should be stated. How were the citations selected? You may describe what keywords were used for the literature search, which databases, and what years covered. How were the HSP inhibitors selected for discussion? Some chemicals that directly interact with HSP are not included, which, for example, dihydromyricetin interacts with HSP78 (GRP78) (https://doi.org/10.1093/jn/nxab057).

Ans: Thank you for your suggestion. In this review, we describe the function and classification of HSPs, the transcriptional mechanism of HSPs promoted by heat shock factors (HSFs), discuss the interaction between HSPs and viruses, and the mode of action of HSP inhibitors at two aspects of inhibiting the expression of HSPs and targeting the HSPs, and elaborate their potential use as antiviral agents. The citations focus on HSPs, HSFs, and how HSP inhibitors affect viruses, with literature search terms such as HSPs, HSFs, HSP inhibitors, etc., from the web of science, years 1970 to 2022. We focus on existing mainstream reported HSP inhibitors, as well as HSP inhibitors that have been screened by studies for clinical trials. We thank you very much for your suggestion, and dihydromyricetin has been added in the revised MS. The revised parts were marked in red in the revised MS. For instance, the abstract of “Following heat stress-induced denaturation, many cellular proteins can regain their function following correct refolding with the aid of molecular chaperones. As a molecular chaperone, heat shock proteins (HSPs) can help client proteins to fold correctly. During viral infection, HSPs are involved with replication, movement, assembly, disassembly, subcellular localization, and transport of the virus via the formation of the macromolecular protein complexes, such as the viral replicase complex. Recent studies have indicated that HSP inhibitors can inhibit viral replication by interfering with the interaction of the virus with the HSP. In this review, we describe the interaction between HSPs and viruses, and discuss the mode of action of HSP inhibitors and their potential use in antiviral activity.” in Page 1, Lines 13-21 of previous MS has been revised to “Mis-folded proteins after stress-induced denaturation can regain their functions through correct re-folding with the aid of molecular chaperones. As a molecular chaperone, heat shock proteins (HSPs) can help client proteins to fold correctly. During viral infection, HSPs are involved with replication, movement, assembly, disassembly, subcellular localization, and transport of the virus via the formation of the macromolecular protein complexes, such as the viral replicase complex. Recent studies have indicated that HSP inhibitors can inhibit viral replication by interfering with the interaction of the virus with the HSP. In this review, we describe the function and classification of HSPs, the transcriptional mechanism of HSPs promoted by heat shock factors (HSFs), discuss the interaction between HSPs and viruses, and the mode of action of HSP inhibitors at two aspects of inhibiting the expression of HSPs and targeting the HSPs, and elaborate their potential use as antiviral agents.” in Page 1, Line 12-22 of revised MS. The paragraph of “6.9. Dihydromyricetin-based HSP70 inhibitors Dihydromyricetin, which is called ampelopsin, is extracted from Ampelopsis grossedentata leaf. Dihydromyricetin can target the ATP binding site of 78-kDa glucose-regulated protein (GRP78) (heat shock 70 kDa protein 5) using the assays of molecular docking and surface plasmon resonance (Figure 7F) [215].” has been added in Page 17-18, Line 614-618 of the revised MS.

1. You may explicitly state that the HSPs you mean in this review are those in the hosts or the viruses.

Ans: Thank you for your suggestion. We have changed the title to “Mode of Action of Heat Shock Protein (HSP) Inhibitors Against Viruses Through Host HSP and Virus Interactions”. The revised parts were marked in red in the revised MS. In Page 1, Lines 2-3 of previous MS, the title of “Mode of Action of Heat Shock Protein (HSP) Inhibitors Against Viruses Through HSP/Virus Interaction” has been revised to the title of “Mode of Action of Heat Shock Protein (HSP) Inhibitors Against Viruses Through Host HSP and Virus Interactions” in Page 1, Line 2-3 of revised MS.

2. L442: Section 6 is the core of this review. Authors may clarify what HSP inhibitors have been specifically identified their direct interaction with HSPs (or HSFs or HSEs) in vitro and in vivo and what methods were used.

Ans: Thank you for your suggestion. We have added details of quercetin as an HSP inhibitor. Such as, the effect of quercetin on cytosolic NbHSP70 mRNA of N. benthamiana was analyzed using northern blot assay, and the effect of quercetin on HSP70 of PepMV-infected N. benthamiana was analyzed using western blot assay, and quercetin could inhibit the binding of HSF to HSE oligonucleotides during heat shock treatment in COLO 320DM cells in vivo. The revised parts were marked in red in the revised MS. In Page 13, Line 464 of previous MS, the words of “using northern blot assay” have been added after the sentence of “cytosolic NbHSP70 mRNA was quickly inhibited to 80% of the level in untreated N. benthamiana leaves.” in Page 13, Line 473 of revised MS; in Page 13, Line 466 of previous MS, the words of “using western blot assay” have been added after the sentence of “the amount of HSP70 following quercetin treatment was reduced by 50– 73% compared with that of the control treatment after 4 days of spraying.” in Page 14, Line 475 of revised MS; in Page 15, Line 504 of previous MS, the words of “in vivo” have been added after the sentence of “quercetin could inhibit the binding of HSF to HSE oligonucleotides during heat shock treatment in COLO 320DM cells.” in Page 15, Line 512 of revised MS.

3. Authors in this section spent large pagination describing quercetin and its analogs. It is well known that flavonoids such as quercetin have low bioavailability. Authors may want to elaborate on their bioavailability.

Ans: Thank you for your suggestion. We have added to the description of the bioavailability of quercetin in the revised MS. The sentences of “Quercetin can inhibit cytokine production, and reduce the activity of cyclooxygenase and lipoxygenase, then exerts powerful anti-inflammatory effects [184-186]. However, due to its poor water solubility, chemical stability, and absorption properties, quercetin's bioavailability is usually less than 10% [187]. At present, the bioavailability of quercetin was enhanced using encapsulation technologies and crystal engineering [188-190].” have been added in Page 13, Lines 453-458 of the revised MS.

4. L2: Title: Perhaps the title can be revised to “Mode of Action of Heat Shock Protein (HSP) Inhibitors Against Viruses”. “Through HSP/Virus Interaction” does not seem to add additional meaning. Or you may have the title as: “Mode of Action of Heat Shock Protein (HSP) Inhibitors Against Viruses through host HSP and Virus Interactions.”

Ans: Thank you for your suggestion. We have changed the title to “Mode of Action of Heat Shock Protein (HSP) Inhibitors Against Viruses Through Host HSP and Virus Interactions”. The revised parts were marked in red in the revised MS. In Page 1, Lines 2-3 of the previous MS, the title of “Mode of Action of Heat Shock Protein (HSP) Inhibitors Against Viruses Through HSP/Virus Interaction” has been revised to the title of “Mode of Action of Heat Shock Protein (HSP) Inhibitors Against Viruses Through Host HSP and Virus Interactions” in Page 1, Line 2-3 of revised MS.

5. L11-12: You may want to revise this sentence. This sentence may be revised to: Molecular chaperones mediate the correct refolding of misfolded proteins after stress-induced denaturation to regain their functions. OR Mis-folded proteins after stress-induced denaturation can regain their functions through correct re-folding with the aid of molecular chaperones.

Ans: Thank you for your suggestion. We have revised to “Mis-folded proteins after stress-induced denaturation can regain their functions through correct re-folding with the aid of molecular chaperones.” The revised parts were marked in red in the revised MS. In Page 1, Lines 11-12 of the previous MS, the sentence of “Following heat stress-induced denaturation, many cellular proteins can regain their function following correct refolding with the aid of molecular chaperones.” has been revised to the sentence of “Mis-folded proteins after stress-induced denaturation can regain their functions through correct re-folding with the aid of molecular chaperones.” in Page 1, Line 12-13 of revised MS.

6. L18-19: … their potential use as antiviral agents.

Ans: Thank you for your suggestion. We apologize for this error in our manuscript. The revised parts were marked in red in the revised MS. In Page 1, Line 18-19 of previous MS, the words of “their potential use in antiviral activity” has been revised to the words of “elaborate their potential use as antiviral agents” in Page 1, Line 21-22 of revised MS.

7. L35: delete “, etc.”.

Ans: Thank you for your suggestion. We apologize for this error in our manuscript. We have removed it in the revised MS. In Page 1, Line 35 of the previous MS, the words of “quercetin and geldanamycin, etc.” has been revised to the words of “quercetin and geldanamycin” in Page 1, Line 38 of revised MS.

8. L45: delete “also”.

Ans: Thank you for your suggestion. We apologize for this error in our manuscript. We have removed it in the revised MS. In Page 2, Line 45 of previous MS, the words of “are also involved with protein degradation” has been revised to the words of “are involved with protein degradation” in Page 2, Line 48 of revised MS.

9. L69” sHSPs possess…

Ans: Thank you for your suggestion. We apologize for this error in our manuscript. The revised parts were marked in red in the revised MS. In Page 2, Line 69 of previous MS, the words of “sHsps possess” have been revised to the words of “sHSPs possess” in Page 2, Line 72 of revised MS.

10. L250 and Figure 3: chemical names can be labeled under each chemical structure.

Ans: Thank you for your suggestion. We have labeled the chemical names under each chemical structure in Figure 3, Figure 5, and Figure 7. In Page 8, Line 249 of previous MS, has been revised to Page 8, Line 251 of revised MS; in Page 14, Line 475 of previous MS, has been revised to Page 14, Line 482 of revised MS; in Page 18, Line 621 of previous MS, has been revised to Page 18, Line 634 of revised MS.

11. L362, Figure 4: define the acronym in Figure 4 in the figure legend.

Ans: Thank you for your suggestion. We have defined the acronym in Figure 4 in the figure legend. The revised parts were marked in red in the revised MS. The sentences of “Abbreviations: HSP, heat shock protein; ER, endoplasmic reticulum; MRC, macromolecular protein complexes; Viral CP, viral coat protein; Viral MP, viral movement protein; Viral RdRp, viral RNA-dependent RNA polymerase; VRC, viral replicase complex.” have been added in Page 11, Line 364-366 of revised MS. In addition, we have defined the acronym in Figure 1, Figure 2, Figure 6, and Figure 8 in the figure legend. The sentence of “Abbreviations: HSF, heat shock factor.” has been added in Page 4, Line 126 of revised MS; The sentence of “Abbreviations: HSP, heat shock protein.” has been added in Page 4, Line 129 of revised MS; The sentences of “Abbreviations: HSP, heat shock protein; HSF, heat shock factor; HSE, Heat shock element.” have been added in Page 15, Line 515 of revised MS; The sentence of “Abbreviations: HSP, heat shock protein.” has been added in Page 15, Line 724 of revised MS.

12. L507, Figure 6: are those direct interactions between quercetin and HSF, HSE, and HSP gene?

 Ans: Thank you for your suggestion. previous studies have shown that quercetin can inhibit HSF transcription, inhibit HSF binding to HSE and HSP genes, and inhibit HSF phosphorylation in mammalian cells. We have modified Figure 6 to improve. Page 15, Line 507 of previous MS, has been revised to Page 15, Line 513 of revised MS.

Reviewer 2 Report

The work "Mode of Action of Heat Shock Protein (HSP) Inhibitors Against Viruses Through HSP/Virus Interaction" is well-prepared describing the structure of the heat shock factors and the regulation of the heat shock proteins and interaction with viruses and the mode of action of HSPs inhibitors as potential use in antiviral activity.

I suggest that you can:

1) Mention briefly the importance of targeting Hsp90 as a potent anticancer agent containing Hsp90 N-terminal ATP binding inhibitors, such as geldanamycin, and its analogues 17AAG and 17DMAG. The therapeutic application of geldanamycin has been limited due to its poor water solubility and severe hepatotoxicity. However, the pharmacokinetic profiles of the analogues are promising in clinical trials.

2) Emphasise quercetin's powerful anti-inflammatory effects, mainly via inhibition of cytokine production such as tumour necrosis factor (TNF)-α, interleukin (IL)-1β, and IL-6 and reduction of cyclooxygenase and lipoxygenase expression (At the introduction in part 6.1).

3) Update some references, because the majority of references date more than 10 years, which negatively impacts the novelty of the topic.

4) Correct the word nucleus in figure 4.

5) Add the legend of the arrows in figure 8 and try to simplify.

I recommend this review be published after minor revision

Author Response

Reviewer 2:

1. Mention briefly the importance of targeting Hsp90 as a potent anticancer agent containing Hsp90 N-terminal ATP binding inhibitors, such as geldanamycin, and its analogues 17AAG and 17DMAG. The therapeutic application of geldanamycin has been limited due to its poor water solubility and severe hepatotoxicity. However, the pharmacokinetic profiles of the analogues are promising in clinical trials.

Ans: Thank you for your suggestion. GA is a molecule with antimicrobial activity or antitumor activity, which has been isolated fromStreptomyces hygroscopicus var. geldanus var. nova. GA and its ana-logues 17-AAG and 17-DMAG can target the N-terminal ATP binding site of Hsp90, then inhibit the proliferation. The therapeutic application of geldanamycin has been limited due to its poor water solubility and severe hepatotoxicity. The revised parts were marked in red in the revised MS. In Page 16, Lines 557-558 of previous MS, the sentence of “GA is an antimicrobial molecule, which has been isolated from Streptomyces hygroscopicus var. geldanus var. nova[204].” has been revised to the sentence of “GA is a molecule with antimicrobial activity or antitumor activity, which has been isolated from Streptomyces hygroscopicus var. geldanus var. nova [204].” in Page 16, Line 564-565 of revised MS. And the sentences fo “GA and its ana-logues 17-AAG and 17-DMAG can target N-terminal ATP binding site of HSP90, then inhibit the proliferation. The therapeutic application of geldanamycin has been limited due to its poor water solubility and severe hepatotoxicity [205].” have been added in Pages 16-17, Lines 565-568 of revised MS.

2. Emphasise quercetin's powerful anti-inflammatory effects, mainly via inhibition of cytokine production such as tumour necrosis factor (TNF)-α, interleukin (IL)-1β, and IL-6 and reduction of cyclooxygenase and lipoxygenase expression (At the introduction in part 6.1).

Ans: Thank you for your suggestion. Quercetin can inhibit cytokine production, and reduce the activity of cyclooxygenase and lipoxygenase, then exerts powerful anti-inflammatory effects. However, due to its poor water solubility, chemical stability, and absorption properties, quercetin bioavailability is usually less than 10%. At present, the bioavailability of quercetin was enhanced using encapsulation technologies and crystal engineering. The revised parts were marked in red in the revised MS. The sentences of “Quercetin can inhibit cytokine production, and reduce the activity of cyclooxygenase and lipoxygenase, then exerts powerful anti-inflammatory effects [184-186]. However, due to its poor water solubility, chemical stability, and absorption properties, quercetin's bioavailability is usually less than 10% [187]. At present, the bioavailability of quercetin was enhanced using encapsulation technologies and crystal engineering [188-190].” have been added in Page 13, Lines 453-458 of revised MS.

3. Update some references, because the majority of references date more than 10 years, which negatively impacts the novelty of the topic.

Ans: Thank you for your suggestion. We have added some references from recent years. The revised parts were marked in red in the revised MS. The reference of “184.  Dower, J.I.; Geleijnse, J.M.; Gijsbers, L.; Schalkwijk, C.; Kromhout, D.; Hollman, P.C. Supplementation of the pure flavonoids epicatechin and quercetin affects some biomarkers of endothelial dysfunction and inflammation in (pre)hypertensive adults: a randomized double-blind, placebo-controlled, crossover trial. J. Nutr. 2015, 145, 1459-1463. https://doi.org/ 10.3945/jn.115.211888.” has been added in Page 28, Lines 1151-1153 of revised MS. The reference of “185.   Murakami, A.; Ashida, H.; Terao, J. Multitargeted cancer prevention by quercetin. Cancer Lett. 2008, 269, 315-325. https://doi.org/10.1016/j.canlet.2008.03.046.” has been added in Page 29, Lines 1154-1155 of revised MS. The reference of “186.      Carullo, G.; Cappello, A.R.; Frattaruolo, L.; Badolato, M.; Armentano, B.; Aiello, F. Quercetin and derivatives: useful tools in inflammation and pain management. Future Med. Chem. 2017, 9, 79-93. https://doi.org/10.4155/fmc-2016-0186.” has been added in Page 29, Lines 1156-1157 of revised MS. The reference of “187.   Cai, X.; Fang, Z.; Dou, J.; Yu, A.; Zhai, G. Bioavailability of quercetin: problems and promises. Curr. Med. Chem. 2013, 20, 2572-2582. https://doi.org/10.2174/09298673113209990120.” has been added in Page 29, Lines 1158-1159 of revised MS. The reference of “188.   Han, L.; Lu, K.; Zhou, S.; Zhang, S.; Xie, F.; Qi, B.; Li, Y. Development of an oil-in-water emulsion stabilized by a black bean protein-based nanocomplex for co-delivery of quercetin and perilla oil. LWT-Food Sci. Technol. 2015, 138, 110644. https://doi.org/10.1016/j.lwt.2020.110644.” has been added in Page 29, Lines 1160-1162 of revised MS.The reference of “189.   Hussein, J.; El-Naggar, M.E. Synthesis of an environmentally quercetin nanoemulsion to ameliorate diabetic-induced cardiotoxicity. Biocatal. Agric. Biotechnol. 2021, 33, 101983. https://doi.org/10.1016/j.bcab.2021.101983.” has been added in Page 29, Line 1163-1164 of revised MS. The reference of “190. Manca, M.L.; Lai, F.; Pireddu, R.; Valenti, D.; Schlich, M.; Pini, E.; Ailuno, G.; Fadda, A.M.; Sinico, C. Impact of nanosizing on dermal delivery and antioxidant activity of quercetin nanocrystals. J. Drug Delivery Sci. Technol. 2020, 55, 101482. https://doi.org/10.1016/j.jddst.2019.101482.” has been added in Page 29, Lines 1165-1167 of revised MS. The reference of “205.  Gorska, M.; Popowska, U.; Sielicka-Dudzin, A.; Kuban-Jankowska, A.; Sawczuk, W.; Knap, N.; Cicero, G.; Wozniak, F. Geldanamycin and its derivatives as Hsp90 inhibitors. Front. Biosci. -Landmark 2012, 17, 2269-2277. https://doi.org/10.2741/4050.” has been added in Page 29, Lines 1203-1204 of revised MS. The reference of “215.     Sun, B.; Tan, D.; Pan, D.; Baker, M.R.; Liang, Z.; Wang, Z.; Lei, J.; Liu, S.; Hu, C.Y.; Li, Q.X. Dihydromyricetin imbues antiadipogenic effects on 3T3-L1 cells via direct interactions with 78-kDa glucose-regulated protein. J. Nutr. 2021, 151, 1717-1725. https://doi.org/10.1093/jn/nxab057.” has been added in Page 30, Lines 1229-1231 of revised MS.

4. Correct the word nucleus in figure 4.

Ans: Thank you for your suggestion. We apologize for this error in our manuscript. We have changed the wrong word to “nucleus” in Figure 4. Please see Page 11, Line 361 of revised MS.

5. Add the legend of the arrows in figure 8 and try to simplify.

Ans: Thank you for your suggestion. We have added the legend of the arrows in figure 8 and tried to simplify. In Page 20, Line 709 of previous MS, has been revised to Page 21, Line 721 of revised MS.

Round 2

Reviewer 1 Report

Thanks to the authors for addressing my comments. Most of my comments have been incorporated into the revised manuscript. However, the following comments have not been addressed.  This review believes that incorporation of the comments can improve the manuscript.  How were the citations selected?  You may describe what keywords were used for the literature search, which databases, and what years covered.  How were the HSP inhibitors selected for discussion?  

Author Response

Reviewer 1:

Thanks to the authors for addressing my comments. Most of my comments have been incorporated into the revised manuscript. However, the following comments have not been addressed. This review believes that incorporation of the comments can improve the manuscript. How were the citations selected? You may describe what keywords were used for the literature search, which databases, and what years covered.  How were the HSP inhibitors selected for discussion?

  Ans: Thank you for your suggestion. In Page 1, Line 36-44 of revised MS, the sentences of “In this review, we searched the literatures from 1970 to 2022 by selecting "HSP", "HSF", "HSP inhibitor", and "antiviral activity" as the keywords in the database of web of science and pubmed. A total of 232 papers were cited in the review. We described the functional mechanisms of interactions between HSPs and viruses. Meanwhile, we elaborated HSP inhibitors, which were mainly divided into two kinds of directly acting on HSPs to affect their function and indirectly acting on HSF to inhibit transcription of HSPs. Some representative HSP inhibitory compounds, such as quercetin, analogues of quercetin, flavonoid compounds, and geldanamycin, etc. were used to reveal the mechanism of HSP regulation.” have been revised.